# An Integrated In Silico and In Vivo Approach to Identify Protective Effects of Palonosetron in Cisplatin-Induced Nephrotoxicity

**DOI:** 10.3390/ph13120480

**Published:** 2020-12-20

**Authors:** Eri Wakai, Yuya Suzumura, Kenji Ikemura, Toshiro Mizuno, Masatoshi Watanabe, Kazuhiko Takeuchi, Yuhei Nishimura

**Affiliations:** 1Department of Integrative Pharmacology, Mie University Graduate School of Medicine, Tsu 514-8507, Japan; 318d026@m.mie-u.ac.jp (E.W.); 317066@m.mie-u.ac.jp (Y.S.); 2Department of Pharmacy, Osaka University Hospital, Suita 565-0871, Japan; ikemurak@hp-drug.med.osaka-u.ac.jp; 3Department of Medical Oncology, Mie University Graduate School of Medicine, Tsu 514-8507, Japan; tomizuno@clin.medic.mie-u.ac.jp; 4Department of Oncologic Pathology, Mie University Graduate School of Medicine, Tsu 514-8507, Japan; mawata@doc.medic.mie-u.ac.jp; 5Department of Otorhinolaryngology—Head and Neck Surgery, Mie University Graduate School of Medicine, Tsu 514-8507, Japan; kazuhiko@clin.medic.mie-u.ac.jp

**Keywords:** cisplatin, nephrotoxicity, drug repositioning, data-driven approach, gene expression signature, adverse events, zebrafish

## Abstract

Cisplatin is widely used to treat various types of cancers, but it is often limited by nephrotoxicity. Here, we employed an integrated in silico and in vivo approach to identify potential treatments for cisplatin-induced nephrotoxicity (CIN). Using publicly available mouse kidney and human kidney organoid transcriptome datasets, we first identified a 208-gene expression signature for CIN and then used the bioinformatics database Cmap and Lincs Unified Environment (CLUE) to identify drugs expected to counter the expression signature for CIN. We also searched the adverse event database, Food and Drug Administration. Adverse Event Reporting System (FAERS), to identify drugs that reduce the reporting odds ratio of developing cisplatin-induced acute kidney injury. Palonosetron, a serotonin type 3 receptor (5-hydroxytryptamine receptor 3 (5-HT3R)) antagonist, was identified by both CLUE and FAERS analyses. Notably, clinical data from 103 patients treated with cisplatin for head and neck cancer revealed that palonosetron was superior to ramosetron in suppressing cisplatin-induced increases in serum creatinine and blood urea nitrogen levels. Moreover, palonosetron significantly increased the survival rate of zebrafish exposed to cisplatin but not to other 5-HT3R antagonists. These results not only suggest that palonosetron can suppress CIN but also support the use of in silico and in vivo approaches in drug repositioning studies.

## 1. Introduction

Cisplatin, an inorganic platinum derivative, is one of the most effective anticancer drugs and is widely used for the treatment of many malignancies, including brain, head and neck, lung, breast, and ovarian cancers [1,2]. Cisplatin acts by crosslinking purine bases in DNA. If the resulting damage cannot be reversed by dedicated DNA repair pathways, the cell undergoes apoptosis [1,2]. However, cisplatin can also damage normal tissues, including the kidney, which may lead to severe adverse effects, such as nephrotoxicity [3]. Cisplatin is imported into renal proximal tubule epithelial cells mainly through organic cation transporter 2, localized on the basolateral membrane, and it is excreted into the urine mainly through multidrug and toxin extrusion protein transporter 1, localized on the apical membrane [4,5,6]. In addition to DNA repair pathways, cisplatin activates other signaling cascades that lead to oxidative stress and inflammation in renal proximal tubule epithelial cells, tubular cell death, decline of glomerular filtration rate, and acute kidney failure [3,7,8]. Cisplatin-induced nephrotoxicity (CIN) is often observed within 14 days after initiation of cisplatin treatment [3]. CIN is generally treated and/or prevented by increasing hydration through the administration of isotonic saline, magnesium supplementation, or by mannitol-induced forced diuresis [9]. The latter approach is considered for patients treated with high doses of cisplatin, but mannitol itself may cause dehydration by over-diuresis [9]. Therefore, there is a pressing need for safer and more effective renoprotective therapies for CIN [3,7,8,10].

In recent years, repurposing of approved drugs for new indications, or drug repositioning, has gained considerable interest as a source of new therapies [11,12,13]. Because the candidate drugs have usually passed phase I safety trials, they can directly enter phase II/III safety/efficacy trials for the new indication, greatly reducing the cost and time associated with novel drug development. Drug repositioning has successfully identified therapies for various diseases, including Parkinson’s disease and dry eye [14], and has been aided by the concept of The Connectivity Map [15,16], which is based on the hypothesis that the mechanism of action of two drugs may be similar if they have similar gene expression signature (GES) and that a drug may be therapeutic if it elicits the opposite GES to that induced by the disease [17,18]. The data-driven approach has previously shown success when applied to drug repositioning [18,19].

In the present study, we sought to identify drugs that might prevent CIN by performing a transcriptome-based drug repositioning study. We identified a GES characteristic of cisplatin-treated kidney cells and then employed bioinformatics tools to search for drugs that might counter the cisplatin-associated GES, thereby preventing CIN. Finally, we validated the effects of one of the identified drugs, palonosetron, a 5-hydroxytryptamine receptor 3 (5-HT3R) antagonist, using an in vivo model of cisplatin-induced toxicity in zebrafish. Our data thus identify a potential novel use for palonosetron in suppressing CIN.

## 2. Results

### 2.1. Identification of a Gene Expression Signature Associated with Cisplatin-Induced Nephrotoxicity

To identify a GES associated with CIN, which we termed GES-CIN, we analyzed transcriptome datasets of cisplatin-exposed mouse kidneys [20,21] and human kidney organoid [22] from the Gene Expression Omnibus [23]. These studies analyzed transcriptome alterations at 2 days [21] or 3 days [20,22] after cisplatin treatment. We identified a GES-CIN of 152 and 56 genes that were significantly upregulated and downregulated, respectively, by cisplatin treatment in all three datasets (Figure 1, Appendix A). To examine the functions related to this expression signature, we performed enrichment analysis using the Kyoto Encyclopedia of Genes and Genomes (KEGG) [24]. The analysis identified significant enrichment of the GES-CIN in a number of signaling pathways and biological functions, including “p53 signaling”, “TNF signaling”, and “tight junction” (Figure 2, Appendix A). Notably, these functions are known to be involved in the mechanisms underlying CIN [7,25], thereby supporting the validity of the 152 and 56 genes identified in this study as a GES-CIN.

### 2.2. Identification of Palonosetron as a Potential Therapeutic Drug for CIN

GESs may be causally related to a particular disease or drug side effect; therefore, it is plausible that drugs that induce the opposite pattern of gene expression to the disease/drug might have therapeutic effects [15]. To identify chemicals that may have a GES opposite to that observed for CIN, we used Cmap and Lincs Unified Environment (CLUE), a database that contains 476,251 expression signatures generated by 27,927 perturbagens, including 19,811 small molecule chemicals [16]. This analysis identified 303 chemicals that were predicted to cause a GES opposite to the GES-CIN (Appendix A). In parallel, we searched Food and Drug Administration Adverse Event Reporting System (FAERS) database [26] to identify drugs that could potentially suppress adverse events caused by other drugs [27,28,29]; in this case, CIN. Using this database, we screened 23,028 patients who experienced acute kidney injury following cisplatin treatment (Appendix A) and identified 16 co-administered drugs that significantly reduced the reporting odds ratio (ROR) of developing acute kidney injury (Appendix A). Palonosetron, a 5-HT3R antagonist in clinical use as an antiemetic [30], was included in the 303 chemicals, identified using CLUE, and the 16 drugs, identified using FAERS. These findings suggested that palonosetron may suppress CIN.

### 2.3. Suppression of CIN in Patients with Head and Neck Cancer by Palonosetron

To test the validity of our hypothesis, we retrospectively analyzed clinical data from 103 patients with head and neck cancers who received cisplatin and 5-fluorouracil for the first time at Mie University Hospital. Among the 103 patients, 26 and 77 patients received the 5-HT3R antagonists ramosetron and palonosetron, respectively, for the treatment of chemotherapy-induced nausea and vomiting. None of the baseline characteristics differed significantly between the patients receiving ramosetron and palonosetron (Table 1).

Analysis of the maximum serum concentrations of creatinine (Scr) and blood urea nitrogen (BUN) in the first 14 days of cisplatin treatment showed that both variables were significantly lower in the palonosetron group compared with the ramosetron group (Figure 3), whereas the overall survival rates of the patients were not significantly different (Figure 4).

### 2.4. Palonosetron Treatment Increases the Survival of Cisplatin-Exposed Zebrafish

Finally, we confirmed the protective effect of palonosetron on cisplatin-induced toxicity using a zebrafish model, in which cisplatin was co-administered with various 5-HT3R antagonists. This analysis revealed that palonosetron, granisetron, and ondansetron did not affect the survival of zebrafish when administered alone; however, palonosetron, but not granisetron or ondansetron, significantly increased the survival rate of zebrafish exposed to a lethal dose of cisplatin (Figure 5). These results suggest not only that palonosetron may suppress cisplatin-induced toxicity but also that the mechanism of action may not be through inhibition of 5-HT3R.

## 3. Discussion

In this study, we performed in silico and in vivo analyses to identify drugs that could be repurposed for the treatment of CIN and found that palonosetron satisfied the search criteria using both CLUE and FAERS tools. We confirmed the renoprotective effects of palonosetron by analysis of data from patients treated with cisplatin and aprepitant, an antagonist of tachykinin receptor 1 (TACR1, also known as neurokinin 1 receptor or substance P receptor) in clinical use as an antiemetic [30] for head and neck cancer, and we verified the in vivo efficacy of palonosetron in suppressing cisplatin-induced toxicity in zebrafish.

Previous work has demonstrated the efficacy of palonosetron combined with netupitant, another antagonist of TACR1, in suppressing cisplatin-induced vomiting in the least shrew (*Cryptotis parva*) compared with either drug alone, and the authors also showed that ERK1/2 phosphorylation in the brain stem was elevated in combination-treated animals [31]. Moreover, palonosetron, but not ondansetron or granisetron, suppressed cisplatin-induced enhancement of neuronal activity in the rat nodose ganglion stimulated by substance P [32]. Aprepitant has been shown to ameliorate CIN in rats, possibly through suppression of oxidative stress [33]. These reports suggest that palonosetron may suppress CIN through inhibition of TACR1 signaling. Palonosetron and netupitant synergistically inhibited somatic and visceral pain in a mouse model of experimental neuropathic pain [34]. Tropisetron, a 5-HT3R antagonist that also inhibits TACR1 signaling [35], suppressed CIN in mice [36]. Although crosstalk between 5-HT3R and TACR1 has been demonstrated [37,38,39], the underlying mechanism and how it is inhibited by palonosetron and tropisetron, but not by ondansetron or granisetron, remain largely unknown. The molecular mechanisms by which palonosetron suppresses CIN also remain to be elucidated.

In the present study, we used the concept of a connectivity map [15,16], a reference database that catalogs GESs in cells exposed to genetic or chemical perturbagens. This approach facilitates the identification of drugs with predicted efficacy for various diseases [15]. However, further analyses are necessary to refine and narrow down drugs identified by GES alone. In the present study, we used CLUE to search for drugs eliciting potential GESs that might counter CIN and, in parallel, we searched FAERS to identify drugs co-administered with cisplatin that potentially suppress CIN. FAERS currently contains >20 million cases of adverse events [26], many of which have been reported in patients receiving multiple drugs concurrently. This analysis yielded 16 co-administered drugs that reduced the ROR for CIN. Of the 303 and 16 drugs identified by CLUE and FAERS, respectively, only palonosetron and paclitaxel were common to both analyses, which facilitated our focus on palonosetron for further analysis.

However, we should note that the concept of connectivity maps and FAERS analyses has some limitations in drug repositioning because the list of candidate drugs identified by these tools is dependent on the submitted GESs, the reference drug perturbation GESs, and the algorithms applied to connect GESs with drugs [18]. First, to generate GES-CIN, we selected three transcriptome datasets that analyzed the effect of 2- or 3-day treatments with cisplatin on the transcriptomes of mouse kidneys and human organoids [20,21]. Although the KEGG analyses confirmed the validity of the GES-CIN identified here, datasets representing the effects of cisplatin on other kidney transcriptomes may yield different results. Second, while we used CLUE in the present study, many other bioinformatic tools using the connectivity map concept employ different reference datasets and algorithms [18]. Third, FAERS compiles reports from various sources, including pharmaceutical companies, doctors, and patients, and the same cases may thus be duplicated or triplicated. Fourth, there is a tendency to observe a higher volume of adverse event reporting during the initial stages of clinical use after drug approval [40]. Ultimately, the list of potential therapeutic drugs generated from FAERS depends on the method of data curation [41]. In this study, we deduplicated cases by selecting the most recent FAERS report and collated drugs by using the 5th level of the Anatomical Therapeutic Chemical classification system. Nevertheless, results predicted in silico should always be validated in vitro and/or in vivo.

In this study, we were able to demonstrate that palonosetron significantly suppressed CIN in patients with head and neck cancer compared to ramosetron. However, we were not able to demonstrate significant differences between palonosetron and ramosetron regarding the dose of cisplatin used for cancer treatment and the overall survival rates. We included 26 and 77 patients, who received ramosetron and palonosetron, respectively, in this retrospective study. Analysis of more patient data will be required to validate the effect of palonosetron in both cancer treatment and CIN in more detail.

Zebrafish have long been used successfully to examine the effects of various chemicals and drugs [42,43,44,45], including CIN, and they have also been used to investigate the potential biological mechanism of CIN [46,47]. For example, zebrafish have been employed to investigate the pharmacological effects of ondansetron and granisetron [48,49] and identify and characterize TACR1 homologs [50], suggesting that this model system can also be used to analyze 5-HT3R and TACR1 signaling. Although we cannot exclude the possibility that palonosetron increased the survival rate of cisplatin-exposed larval zebrafish through mechanisms other than renoprotection, our preliminary studies using adult zebrafish suggest that palonosetron suppresses kidney injury molecule 1 expression induced by cisplatin exposure (data not shown).

In summary, we employed an in silico approach to identify palonosetron as a potential therapy for CIN and validated the protective effects in vivo using zebrafish. 5-HT3R antagonists have been widely used to treat cisplatin-induced nausea and vomiting, and our study results suggest that palonosetron may also protect against CIN. Further clinical studies will be required to confirm this hypothesis. Our study also suggests that the integrated approach of in silico prediction and in vivo validation using zebrafish may be applicable to drug repositioning for other disorders.

## 4. Materials and Methods

### 4.1. Ethics Statement

This study was conducted in accordance with the Declaration of Helsinki and was approved by the Ethics Committee of Mie University Graduate School of Medicine and Faculty of Medicine (no. H2019-038). Informed consent was obtained from participants through an opt-out method because the data were collected retrospectively from electronic medical records. Mie University Institutional Animal Care and Use Committee guidelines state that no approval of the study protocol is required for experiments using zebrafish. However, all animal experiments described in this manuscript conform to the ethical guidelines established by the Institutional Animal Care and Use Committee at Mie University.

### 4.2. Transcriptome Analysis

Datasets were obtained from Gene Expression Omnibus [23]. We used Galaxy [51] to convert SRA files in the GSE106993 (SRX3398935-8 and SRX3398943-6 for cisplatin and control groups, respectively), GSE130814 (SRX5801653-54 and SRX5801649-50 for cisplatin and control groups, respectively), and GSE145085 (SRX7707233-6 for cisplatin group and SRX7707231-2 and SRX7707237-8 for control groups) to FASTQ files, to map the reads to mouse or human genomes and to count the reads per gene. We used the R/Bioconductor TCC package [52] to identify genes differentially expressed between control and cisplatin-treated groups using a false discovery rate threshold of 1%. The mouse gene symbols were converted to the human orthologs using DAVID [53]. ClueGO [24] in Cytoscape [54] was used to identify functional pathways for GES-CIN genes in KEGG [55].

### 4.3. Bio/Chemoinformatics Analysis

To identify chemicals that potentially induce a GES opposite to GES-CIN, we employed CLUE (https://clue.io/), a tool that catalogs the GES of human cells subjected to chemical and genetic perturbation. The 152 and 56 GES-CIN genes (Appendix A) were uploaded to CLUE, and 303 chemicals with negative tau scores [16] were identified as potentially inducing the opposite GES to that of GES-CIN. To identify drugs in clinical use that might suppress CIN, we downloaded 6,626,502 files consisting of 10,600 drugs and 18,934 adverse reactions recorded between 2015 and 2019 in FAERS (https://www.fda.gov/drugs/questions-and-answers-fdas-adverse-event-reporting-system-faers/fda-adverse-event-reporting-system-faers-latest-quarterly-data-files). We constructed a GUI-based calculator using Python to analyze the effect of a co-administered drug (d) on the ROR for an adverse reaction (r) reported in patients treated with a drug causative for the adverse event (drug A). The ROR was calculated from two-by-two contingency tables comprised of the number of drug A-treated cases who received d and had r (d1r1), received d and did not have r (d1r0), did not receive d and had r (d0r1), and did not receive d and did not have r (d0r0). The ROR was calculated with 95% confidence intervals. Drugs with maximum 95% confidence intervals <1 were identified as candidate drugs that might suppress CIN.

### 4.4. Analysis of Electronic Medical Records

Clinical data were collected from electronic medical records of 103 hospitalized patients who received cisplatin and 5-fluorouracil treatment for the first time for head and neck cancer at Mie University Hospital between January 2010 and December 2018. Eligible patients received a continuous infusion of 5-fluorouracil (800 mg/m^2^) for 5 days and a 2-h intravenous infusion of cisplatin (80 mg/m^2^). For prevention of nephrotoxicity, all eligible patients received hydration therapy with mannitol. We excluded patients if their serum Scr level and BUN were >1.2 and >21 mg/dL, respectively, before cisplatin treatment and if they were not receiving aprepitant for the treatment of nausea and vomiting. We compared serum Scr and BUN values obtained before and after cisplatin treatment in patients who received ramosetron (N = 26) or palonosetron (N = 77) plus aprepitant for emesis.

### 4.5. Zebrafish Experiments

An albino zebrafish line [56] was used to examine the effect of 5-HT3R antagonists on cisplatin-induced toxicity. Zebrafish were maintained as described previously [57]. Briefly, zebrafish were raised at 28.5 ± 0.5 °C with a 14 h/10 h light/dark cycle. Embryos were obtained via natural mating and were cultured in 0.3× Danieau’s solution (19.3 mM NaCl, 0.23 mM KCl, 0.13 mM MgSO_4_, 0.2 mM Ca(NO_3_)_2_, 1.7 mM HEPES, pH 7.2). At 5 days post-fertilization, zebrafish were exposed for 24 h to final concentrations of 1 mM cisplatin (Tokyo Chemical Industry, Tokyo, Japan) with or without 20 μM palonosetron (Cayman Chemical, Ann Arbor, MI, USA), 20 μM granisetron (Tokyo Chemical Industry, Tokyo, Japan), or 20 μM ondansetron (Tokyo Chemical Industry, Tokyo, Japan), all dissolved and diluted in 0.3× Danieau’s solution.

### 4.6. Statistical Analyses

Statistical analysis was performed using Prism 6 (GraphPad, San Diego, CA, USA). Differences between two groups were analyzed by the Mann–Whitney U test and Fisher’s exact test for continuous and categorical variables, respectively. Survival rate of patients was analyzed using the Kaplan–Meier method and log-rank test. Survival rate of zebrafish was analyzed using the Dunnett’s test. A *p* value of <0.05 was considered significant.

## Figures and Tables

**Figure 1 pharmaceuticals-13-00480-f001:**
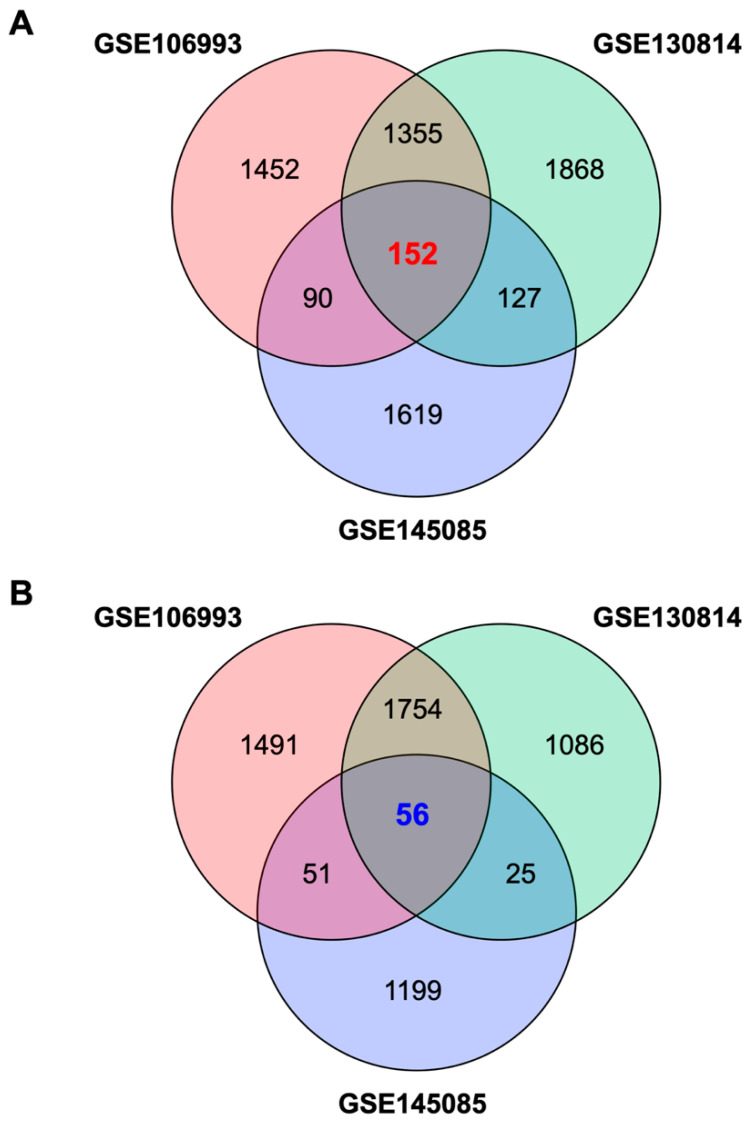
Venn diagrams of genes contained in the gene expression signature associated with cisplatin-induced nephrotoxicity (GES-CIN). Venn diagrams of unique and shared differentially expressed genes in untreated vs. cisplatin-treated mouse kidneys (GSE106993 and GSE130814) or human kidney organoids (GSE145085). Differentially expressed genes were identified using a false discovery rate threshold of 1% (Appendix A). (**A**,**B**) show genes upregulated and downregulated by cisplatin treatment, respectively.

**Figure 2 pharmaceuticals-13-00480-f002:**
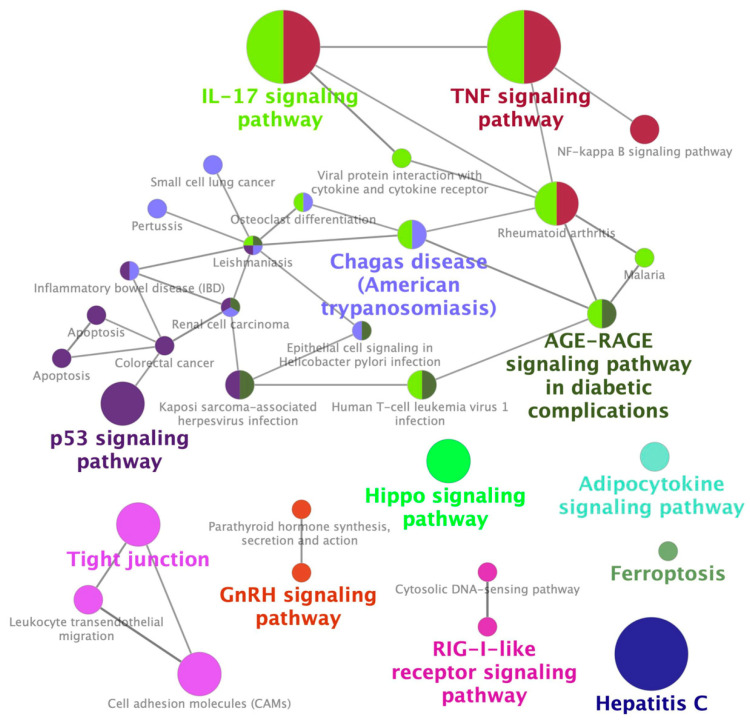
Enrichment of CIN-associated differentially expressed genes in Kyoto Encyclopedia of Genes and Genomes (KEGG) pathways. The 208 genes in the GES-CIN (Appendix A) were analyzed with ClueGO using KEGG as the reference database to identify significantly enriched functional networks.

**Figure 3 pharmaceuticals-13-00480-f003:**
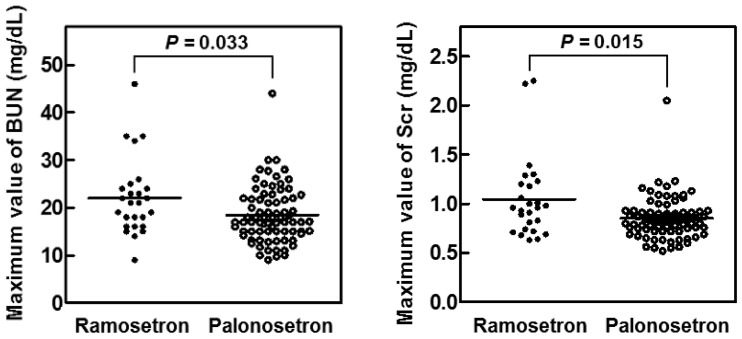
Maximum values of serum creatinine (Scr) and blood urea nitrogen (BUN) in patients treated for 14 days with cisplatin and either ramosetron or palonosetron. Patients with head and neck cancer (*n* = 103) received cisplatin and 5-fluorouracil together with ramosetron (*n* = 26) or palonosetron (*n* = 77). *p* values determined by the Mann–Whitney U test.

**Figure 4 pharmaceuticals-13-00480-f004:**
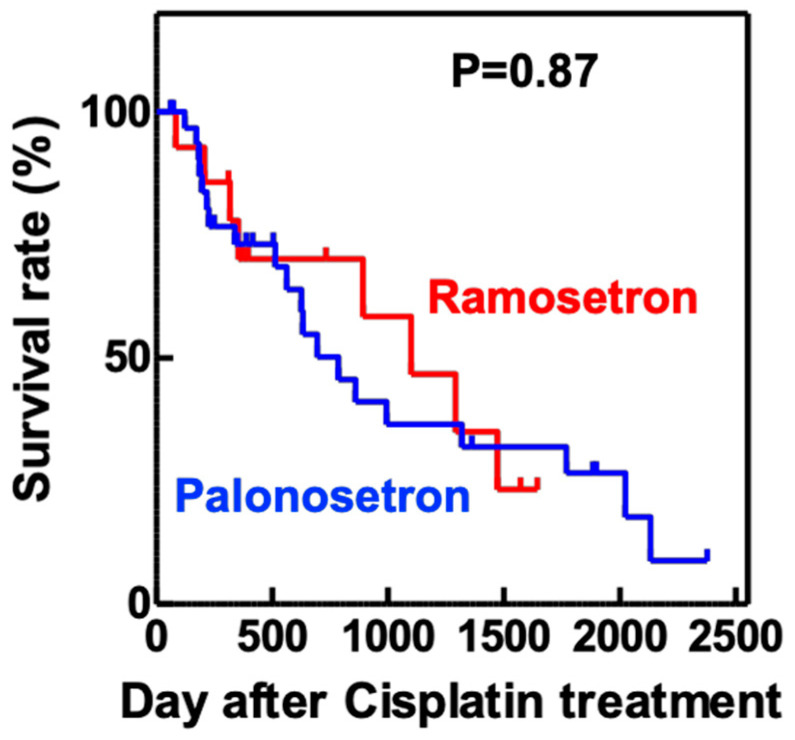
Kaplan–Meier overall survival analysis of patients treated with cisplatin and either ramosetron or palonosetron. Patients with head and neck cancer (*n* = 103) received cisplatin and 5-fluorouracil, together with either ramosetron (*n* = 26) or palonosetron (*n* = 77), to treat chemotherapy-induced nausea and vomiting. *p* value determined using the log-rank test.

**Figure 5 pharmaceuticals-13-00480-f005:**
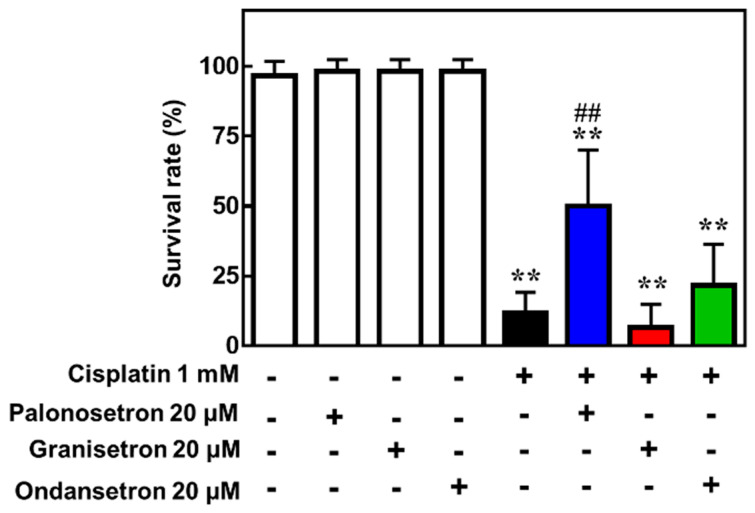
Effects of 5-hydroxytryptamine receptor 3 (5-HT3R) antagonists on the survival of zebrafish exposed to cisplatin. Zebrafish (5 days post-fertilization) were treated with the indicated combinations of drugs for 24 h, and survival was assessed. Mean ± SEM of *n* = 6. ** *p* < 0.01 vs. no drug group, ## *p* < 0.01 vs. cisplatin only group by Dunnett’s test.

**Table 1 pharmaceuticals-13-00480-t001:** Characteristics of patients treated with ramosetron or palonosetron.

Patients’ Characteristics	All Patients (*n* = 103)	Ramosetron (*n* = 26)	Palonosetron (*n* = 77)	*p* Value
Male	88 (85)	22 (85)	66 (86)	0.891
Age (years)	64 (33–78)	66 (33–78)	64 (34–76)	0.111
Body weight (kg)	52 (34–85)	52 (34–72)	51 (34–85)	0.499
Body surface area (m^2^)	1.56 (1.29–2.02)	1.57 (1.29–1.85)	1.56 (1.22–2.02)	0.463
Smoking history	86 (83)	21 (81)	65 (84)	0.606
Drinking history	67 (65)	17 (65)	50 (65)	0.995
5-FU dose (mg/day)	1260 (900–1640)	1225 (1000–1500)	1270 (900–1640)	0.204
Cisplatin dose (mg/day)	125 (80–164)	120 (95–150)	125 (80–164)	0.105
Baseline biological parameters				
BUN (mg/dL)	11.0 (6.5–19.0)	12.0 (7.0–19.0)	11 (6.5–19.0)	0.445
Scr (mg/dL)	0.73 (0.44–1.02)	0.77 (0.45–1.01)	0.72 (0.44–1.02)	0.538
Hemoglobin (g/dL)	12.4 (8.2–16.9)	12.1 (8.9–15.1)	12.5 (8.2–16.9)	0.463
Platelet (×10^9^ L)	220 (58–417)	217 (82–407)	220 (58–417)	0.566
White blood cells (×10^9^ L)	5.27 (2.10–14.9)	6.20 (2.10–14.9)	5.14 (2.21–14.4)	0.477
Co-administrated				
NSAIDs	12 (12)	4 (15)	8 (10)	0.396
Magnesium oxide	26 (25)	7 (27)	19 (25)	0.776
Proton pump inhibitors	14 (14)	4 (15)	10 (13)	0.632

5-fluorouracil (5-FU); blood urea nitrogen (BUN); serum concentrations of creatinine (Scr); non-steroidal anti-inflammatory drugs (NSAIDs).

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
