# Peer review of "An Integrated In Silico and In Vivo Approach to Identify Protective Effects of Palonosetron in Cisplatin-Induced Nephrotoxicity"

_pharmaceuticals, 2020, doi:10.3390/ph13120480_

Round 1

Reviewer 1 Report

THe paper is very interesting and well described.

I Have only the minor suggestions

  1. Please give the full name od CLUE in the abstract.
  2. please give the name and version od the statistical softwer.

Author Response

Reviewer 1

The paper is very interesting and well described.

I have only the minor suggestions.

Please give the full name of CLUE in the abstract.

please give the name and version of the statistical software.

Thank you very much for your positive feedback.

We have revised the manuscript according to the comments (line 25 and line 374).

Reviewer 2 Report

As a result of the study, the expression profiles of genes that change during therapy with cisplatin are shown. The hypothesis is checked with help of the initial retrospective material. Authors suggest that by using the transcriptome maps it is possible to prevent the nephrotoxicity of cisplatin. 

To improve understanding of the article, it is necessary to describe the introduction in more detail and add the significance of transcriptome analysis. Why, at this stage, a search for potential targets for reducing toxicity was carried out using only databases. I would also like the authors to be more specific about the importance of preliminary retrospective studies in this article. Please modify the introduction and conclusion part of the paper.

Author Response

Reviewer 2

As a result of the study, the expression profiles of genes that change during therapy with cisplatin are shown. The hypothesis is checked with help of the initial retrospective material. Authors suggest that by using the transcriptome maps it is possible to prevent the nephrotoxicity of cisplatin. To improve understanding of the article, it is necessary to describe the introduction in more detail and add the significance of transcriptome analysis. Why, at this stage, a search for potential targets for reducing toxicity was carried out using only databases. I would also like the authors to be more specific about the importance of preliminary retrospective studies in this article. Please modify the introduction and conclusion part of the paper.

Thank you very much for your constructive comments.

We have revised the manuscript to clarify the significance of transcriptome analysis and data-driven approach.

“Drug repositioning has been aided by the concept of The Connectivity Map, which is based on the hypothesis that the mechanism of action of two drugs may be similar if they have similar gene expression signature (GES), and that a drug may be therapeutic if it elicits the opposite GES to that induced by the disease. The data-driven approach has previously shown success when applied to drug repositioning.” (lines 65-71).

We have also revised the manuscript to be more specific about the importance of preliminary retrospective studies.

“In this study, we were able to demonstrate that palonosetron significantly suppressed CIN in patients with head and neck cancer compared to ramosetron. However, we were not able to demonstrate significant differences between palonosetron and ramosetron regarding the dose of cisplatin used for cancer treatment and the overall survival rates. We included 26 and 77 patients who received ramosetron and palonosetron, respectively, in this retrospective study. Analysis of more patient data will be required to validate the effect of palonosetron in both cancer treatment and CIN in more detail.” (lines 281-287).